# Scalable Range Search over Temporal and Numerical Expressions

## Abstract

Natural language expressions of time and numbers can be ambiguous (e.g., *2020s* can refer to either 2021 or 2025), can be present at different granularities, or can be unbounded (e.g., *more than ten percent*). To match and retrieve such ambiguous temporal and numerical expressions over millions of documents, we present NASH. Our experiments on collections amounting to more than 22 million documents show that NASH provides significant speedups in the order of 19.23 - 53.10× for `contains` and `near` queries. NASH manages this while using indexes that are 1.90 - 2.05× smaller than the indexes utilized by baselines. We further demonstrate NASH's scalability to the Web by indexing a subset of Common Crawl amounting to more than 365 million documents.

## CCS Concepts

• **Information systems → Search engine indexing**.

## Keywords

Indexing, Efficiency, Analytics, Temporal & Numerical Expressions

**ACM Reference Format:**
Anonymous Author(s). 2024. Scalable Range Search over Temporal and Numerical Expressions. In *Proceedings of ICTIR'24.* ACM, New York, NY, USA, 10 pages. https://doi.org/10.1145/1122445.1122456

## 1 Introduction

The Web is a knowledge-rich repository of real-world events as they have unfolded over time. Event analytic tasks require retrieval of documents containing combinations of words, entities, temporal and numerical expressions. Important events are marked by the days, months, or years (e.g., *recession in '24*) which they spanned and are often characterized quantitatively by numbers (e.g., *10% inflation*). To retrieve documents that contain time intervals and numerical values that are of relevance to the ones mentioned in the query is challenging. The first challenge is of performing a semantic match between the queried interval and the intervals described by the temporal and numerical expressions in text (e.g., matching the ambiguous expression *around ten percent* to the query term *11%*). The second challenge is that of the design of a data model such that we can store the values of the temporal and numerical expressions with their precise semantics so as to accommodate their hierarchical (e.g., day < month < year) and proximate relationships. The third and final challenge we face is that of performing the semantic match at scale across

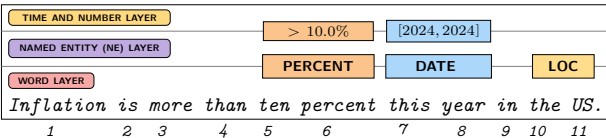

**Figure 1: The annotated text model [18, 19].**

millions of documents. Specifically, the number of implicit intervals referred by *2000s* or *around 10%* can be arbitrarily large, how can we match the time interval expressed in the query to these numerous variations during query processing and avoid inspecting irrelevant intervals (e.g., by naïvely scanning an interval index over the begin and end of the temporal and numerical expressions)?

NASH enables querying for temporal and numerical intervals using range-based operators (i.e., `contains`, `containedBy`, `intersect`, and `near`). To do so, NASH models ambiguous temporal or numerical expression using two-dimensional geometry to overcome the challenge of semantic gap. To store these two-dimensional geometries associated with temporal and numerical expressions, NASH utilizes z-order curves [26]. By utilizing z-order curves, we can query temporal and numerical intervals at different levels of granularity by simply truncating bits of the hash values. Finally, we describe the concept of continuous hashes and an improved variant of BIGMIN and LITMAX method [34] to optimize query processing for NASH.

## 2 NASH

We now describe our system NASH in detail.

### 2.1 Preliminaries

**Annotated Text Model.** We build upon the text model [18, 19], where each document $d$ in a collection $D$ is modeled as a collection of annotation layers: $d_{\mathcal{L}} = \langle \ell_{[i,j]} \dots \ell_{[p,q]} \rangle$. The annotations $\ell$ are drawn from the annotator's (e.g., a named entity tagger) vocabulary $\Sigma_{\mathcal{L}}$ (e.g., PERSON, ORGANIZATION, LOCATION, DATE, NUMBER). Furthermore, each annotation $\ell_{[i,j]}$ adorns a contiguous word sequence $\langle w_{[i,i]}, \dots, w_{[j,j]} \rangle$ in the word layer $d_{\mathcal{W}}$ (see Figure 1).

**Temporal and Numerical Expressions** in text can be obtained by annotators such as Stanford CoreNLP [25]. Specifically, coarse-grained named entity annotations indicated by TIME, DATE, MONEY, NUMBER, PERCENT, ORDINAL can be resolved to precise intervals with the help of rule-based taggers, such as SUTime [13]. Thus, we can resolve implicit, relative, and explicit mentions of time and numbers to exact intervals based on document metadata or explicit dates or numbers mentioned elsewhere in the document contents. Note that unlike temporal expressions, resolved values for numerical expressions can be unbounded (e.g., $> 10\%$) or approximate (e.g., $\sim 5$).

**Uncertainty-Aware Model.** Temporal and numerical expressions in text (e.g., *around millions*) often convey an interval $[b, e]$ which is ambiguous. The uncertainty-aware model [11] represents this ambiguity by allowing for lower and upper bounds to the beginning and the end of intervals. Thus, we can model uncertainty as a quadruple: $\langle b_l, b_u, e_l, e_u \rangle$, where, the lower $l$ and upper $u$ bounds allow ambiguity in the beginning $b$ and end $e$ of the interval ($b \leq e$). Thus, a temporal or numerical interval (e.g., early '20s) can

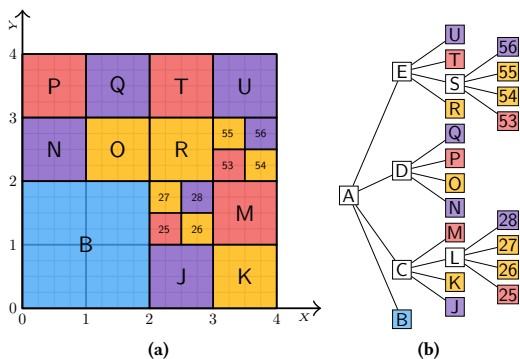

**Figure 2: (a) Modeling data in two-dimensional Cartesian plane, its corresponding quad-regions, and (b) the corresponding quadtree.**

be represented as a two-dimensional bounding box in a Cartesian plane as shown in Figure 3a and refer to its intervals (e.g., [1922, 1925]) as a region in the Cartesian plane.

**Space-Filling Curves** provide a way of linearizing multidimensional values by traversing every point of a $n$-dimensional region and mapping the multidimensional data to a position on an one-dimensional curve. Thus, any $n$-dimensional point can be described by its unique 1-dimensional position on a linear curve. In this work, our focus will be on space-filling curves and their applications in two-dimensional space.

**The Quadtree** [17] is a common technique used for spatial indexing. Quadtrees share some characteristics with space-filling curves, such as: the space division into regions and visualization of the hierarchical characteristics as a tree. The quadtree is a generalization of the binary tree for treating data in two dimensions. A binary tree recursively splits a one-dimensional space in two, resulting in a maximum of $2^n$ nodes at each level, starting with $n = 0$ at the top level. The quadtree functions similarly but in two dimensions. Each dimension is split in two, resulting in four total children and thus $4^n$ nodes of each level. The region quadtree is a variant of the quadtree where the space is divided into equal size regions. Each region then represents a two-dimensional range, i.e., a minimum bounding rectangle. All points in the two dimensional space will then belong to one of these regions. Figures 2a and 2b show how a two-dimensional space can be broken into smaller quadrants with the corresponding tree-representation. The tree-representation also illustrates the hierarchical characteristic of the quadtree. The parent-child relationship of nodes means that all child nodes of the same parent node are all contained by the parent region.

## 2.2 Query Language

To retrieve documents for event-centric queries, we next explain the range-based operators that are part of NASH's query language.

**Range-Based Operators.** NASH provides three important range-based operators: `contains` and `containedBy`; `intersect`; and `near`. We next define the semantics of each of the operators formally. The `contains` operator allows for retrieval of those documents that have an interval which lies completely within the queried interval. Another variation of the operator is `containedBy` which retrieves those documents having intervals which completely subsume the queried interval. The semantics of `contains` (semantics of `containedBy` can be derived similarly) for a queried temporal or a

numerical interval $[b, e]$ can be defined as:

$$\texttt{contains}(\ell) = \left\{ d \in D \mid (\ell' \in d_{\mathcal{L}}) \wedge (\ell' \cap \ell = \ell') \right\}. \quad (1)$$

The `intersect` operator as the name suggests would retrieve those documents that contain a temporal or numerical annotation whose interval representation intersects with the queried interval. The semantics of `intersect` operator can be formalized as:

$$\texttt{intersect}(\ell) = \left\{ d \in D \mid (\ell' \in d_{\mathcal{L}}) \wedge (\ell' \cap \ell \neq \emptyset) \right\}. \quad (2)$$

The final operator `near` retrieves those documents that contain a temporal or numerical annotation whose interval representation is in proximity to the query interval. The proximity function (e.g., Euclidean distance) is defined during indexing of the document collection and allows us to retrieve those documents quickly within a given margin $\Delta$. The semantics of the `near` operator is defined as:

$$\texttt{near}(\ell) = \left\{ d \in D \mid (\ell' \in d_{\mathcal{L}}) \wedge (\texttt{distance}(\ell, \ell') \leq \Delta) \right\}. \quad (3)$$

## 2.3 Index Design

A naïve approach to searching temporal and numerical intervals would be to index each dimension of the interval (e.g., begin and end) and evaluate the range-based search predicates. However, either with a naïve interval representation or the uncertainty-aware interval representation this would require multiple scans over the indexes to determine the result set. An alternative approach can be to utilize known indexing techniques for multidimensional data [33]. However, these approaches are more involved and hard to accommodate in the query processing engine of a large scale search infrastructure that primarily relies on inverted indexes. To bring together a unified indexing solution to annotated text documents we turn to space-filling curves, which allow us to map multidimensional data into one-dimensional summaries that can then be queried using inverted indexes. Space-filling curves also exhibit locality-preservation property such that geometries or data points in high dimensional spaces that are close together are allocated proximate hash values. This provides data locality on disk thereby aiding in quick query processing. Furthermore, space-filling curves inherently provide hierarchical relationship between coarser-granular regions and finer-granular regions (similar to QuadTrees for indexing spatial data [17]). This property of space-filling curve is particularly important as it also allows us to capture the hierarchical relationship in temporal expressions mentioned at different granularities (i.e., day < month < year). For our purposes, as temporal and numerical intervals are represented in the uncertainty-aware representation, we are concerned with mapping two-dimensional bounding boxes to one-dimensional summaries.

**z-order Curves.** We utilize z-order curves [26] for computing the hash values. Other space-filling curves such as Hilbert curves [21] allow for higher precision and better locality preservation but at an increased computation cost. A z-order curve partitions the Cartesian space recursively into four disjoint regions. To assign a unique value to each of these regions the curve traverses through each of them exactly once in a "z" manner. The z-order hash for a region can be arrived by interleaving the bits representing the coordinate positions spanning the partitions (see Figure 3b).

Using the z-order curve, we can now model the uncertainty-aware representation of temporal and numerical expressions. This

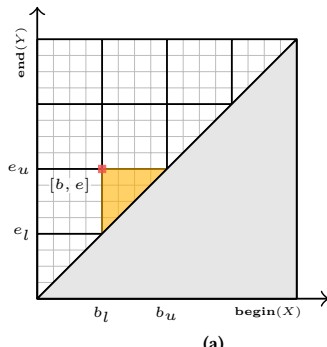

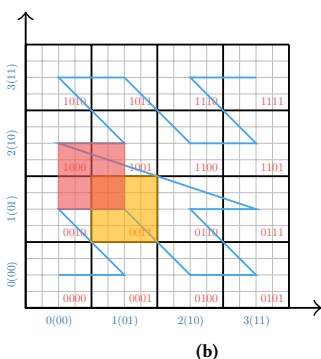

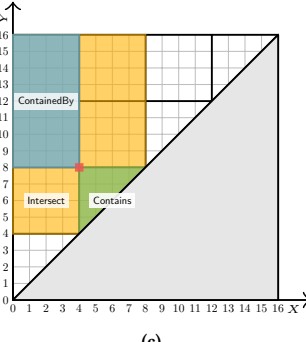

| (a) | (b) | (c) |

**Figure 3: (a) The uncertainty-aware model with an example temporal expression (yellow region) and an interval $[b, e]$. (b) z-order hashes for two temporal expressions. (c) Region-based query operations translated as MBRs for the queried red MBR.**

```
Input:  X: X-coordinate, Y: Y-coordinate, XR: range of X, YR: range of Y, p: precision.
Output:  h: the calculated z-order value.
 1: procedure calculateHash(X, Y, XR, YR, p)
 2:     h ← bit string with zero-bits, length of p
 3:     b ← 0                                              ▷ Current bit position
 4:     while b < p do
 5:         if b is even then                              ▷ Y-dimension
 6:             mid ← (YR[0] + YR[1]) / 2
 7:             if Y ≥ mid then
 8:                 YR[0] = mid
 9:                 h ← set current bit of h to 1
10:             else
11:                 YR[1] = mid
12:         else                                           ▷ X-dimension
13:             mid ← (XR[0] + XR[1]) / 2
14:             if X ≥ mid then
15:                 XR[0] = mid
16:                 h ← set current bit of h to 1
17:             else
18:                 XR[1] = mid
19:         // next bit
20:         b ← b + 1
21:         h ← left shift h by 1
22:     return h
```

**Algorithm 1: z-order Hash Computation.**

is done by breaking down the bounding-box corresponding to the expression and computing z-order hashes for the resulting regions. The accuracy of the z-order hash representation of the bounding-box is given by the size of the smallest possible region. The size of a region can be computed by the number of bits and the size of the dimensions being represented. For a range $[b, e]$ and $n$ bits to describe one dimension, accuracy can be defined as:

$$\text{Accuracy} = (e - b)/2^n. \tag{4}$$

**z-order Hash Computation** is shown in Algorithm 1. The `calculateHash` method takes as its input the co-ordinates for which the z-order hash needs to be computed, the overall range for temporal or numerical expression in the entire document collection, and the required precision of the hash. The overall range is defined by an interval, where the begin is the earliest time point or the smallest numerical value while the end is the latest time point or the largest numerical value to occur in the entire document collection. The z-order value is calculated one bit at a time, starting with the most significant bit (MSB). This represents the first Y-value of the hash. A counter $b$ is used to indicate the current bit position. If the position is even, the current bit represents a part of the $Y$-value, and if not, it represents a part of the $X$-value. The calculation of the current bit is the same for both dimensions. The mid-point of the current

dimension is calculated and compared against the mid-point of the current range for that dimension. It starts as the $X$ and $Y$ dimension bounds and is halved for each bit being calculated. If the value is larger than or equal to the midpoint, the current bit position is set to 1, and if not, it stays at zero. The range of the current dimension is then halved, keeping the half containing the value. This is done alternately for the $X$ and $Y$ dimension, until the desired number of bits or precision is reached.

Decoding of the z-order hash is done in a similar manner. Starting with the MSB, if the bit is a 1, the overall range of the dimension (starting with Y) is split into the upper half, and if the bit is 0 into the lower half. This is done for each bit, each time halving the range for the current dimension being examined. The result is then an interval for each dimension, indicating the region the z-order hash covers.

As the range search over the z-order curve is done at a minimum of two bits at a time, a base 4 encoding is used. That is, each base 4 character represents one partition in both the X and the Y dimension. This results in a shorter representation while still allowing for the finest precision increment or reduction by adding or removing from the hash. Moreover, it allows for any even level of z-order precision while indexing and querying. The encoding is as simple as translating each $X$, $Y$ pair into its base 10 representation: $00_2 \rightarrow 0_{10}, 01_2 \rightarrow 1_{10}, 10_2 \rightarrow 2_{10}, 11_2 \rightarrow 3_{10}$.

**z-order Hash Indexes.** To leverage the benefits of the space-filling curves outlined earlier, we instantiate an inverted index that stores the z-order hash along with their positions. The positional information can further help us to compute the shorter text regions within documents that can contain them in sequence with other annotations (e.g., named entities). With this index design choice, we leverage the locality-preservation property of the z-order curves. This is because the points close in the uncertainty-aware model are also stored close together in the index (since the indexes are sorted by the indexing units). Furthermore, we additionally index the prefixes of each of the z-order hashes to leverage the hierarchical property of the space-filling curves. For example, for the hash 3300030030 corresponding to the range $[1890, 1890]$, each prefix 330003003, 33000300, 3300030, etc. are indexed at the same positional span. Therefore, by truncating bits from a hash we can move from fine granular representation of the interval (e.g., at day-level) to a coarse-granular representation of the interval (e.g., at year-level). Concretely, z-order hash indexes store for each occurrence of

| 3300030 | : | 1 | 4 | ⟨1, 2, 3, 3⟩ | ⟨2, 7, 10, 15⟩ | ⟨3, 8, 11, 16⟩ |
|---------|---|---|---|---|---|---|

INDEXING UNIT · D-ID · FREQ. · SENTENCE IDS · BEGIN POSITIONS · END POSITIONS

**Figure 4: The inverted index organization.**

the hash and its prefixes, the following particulars in the document collection: the document identifiers, the sentence identifiers, the begin positions in the document, and the end positions in the document (see Figure 4). The sentence identifiers and positional information allows us to further compute concise text regions containing temporal and numerical expressions in combination with other word sequences and coarse-grained named entities (e.g., PERSON, ORGANIZATION, or LOCATION). The payloads for each of the hashes are further compressed using patched frame of reference [6, 38].

**Index Combinations** are often leveraged for Web-scale query processing [18, 28, 36]. To evaluate event-oriented queries, we additionally require indexes over n-grams (unigrams, bigrams, and trigrams) and coarse-grained named entities (e.g., PERSON, ORG., and LOC.). To construct these indexes, we rely on the same index structure as that of the z-order hash indexes.

## 2.4 Query Processing

Using the z-order hash-based indexes we can now convert search over the uncertainty-aware model into a simple linear range search over the hashes. This naïve approach entails, creating the query-MBR (minimum bounding rectangle) that describes the interval to be searched having been represented in the uncertainty-aware model. The resulting matches can then be obtained by retrieving all the z-order hashes that lie between the diagonally opposite hashes describing the query-MBR. Due to the nature in which the z-order curve jumps across partitions, there are bound to be some false-positives. However, we are guaranteed to always retrieve all of the relevant temporal or numerical expressions with this naïve approach. We next describe the query-MBR generation for the four query predicates (see also Figure 3c).

**Query MBRs for contains and containedBy.** A contains query translates to the MBR described by $X \in [b_x, e_x]$ and $Y \in [b_y, e_y]$ with $X \leq Y$. The result set can be obtained by retrieving all the hashes that lie between diagonally opposing corners of the MBR, i.e., $(b_x, b_y)$ and $(e_x, e_y)$. A containedBy query translates to the MBR described by $X \leq b$ and $Y \geq e$, where the lower bound for $X$ and upper bound for $Y$ is equal to the lower bound and upper bound for the overall range of temporal or numerical annotations being indexed for the document collection, respectively.

**Query MBRs for intersect.** A intersect query translates to a MBR described by $X, Y \in [b, e]$ with $X \leq e, Y \geq b$, where the lower bound for $X$ and upper bound for $Y$ is equal to the lower bound and upper bound for the overall range of temporal or numerical annotations being indexed for the document collection, respectively. Thus, a intersect query includes results that would be obtained from a containedBy and contains query with additional results that intersect the queried interval.

**Query MBRs for near.** We use the Euclidean distance as a measure of proximity to process a near query. A near query translates to a MBR described by the points $[b - \Delta, e - \Delta]$ and $[b + \Delta, e + \Delta]$ as being the diagonally opposing points. In that MBR, $\Delta$ denotes the distance within which an interval represented by a temporal or a numerical expression is seen as being proximate to the queried interval. With near query we can obtain those expressions of time

---

**Input:** $T_1 = [b_x, e_x]$ and $T_2 = [b_y, e_y]$: MBR point tuples, H (=∅): list of hashes, $GR$: tuple of the overall range in D, and p: precision.
**Output:** H: minimal set of hashes covering the input MBR.
1: **procedure** rangeSearch(T₁,T₂, H, GR, p)
2:   cb ← number of common MSBs between T₁ and T₂
3:   **if** T₁ and T₂ are continuous **then**                    ▷ range within MBR
4:     h ← bit string equal to the common MSBs
5:     H.add(h)                                              ▷ hash shorter than p
6:     **return** H
7:   **if** cb = p **then**                                       ▷ at lowest level
8:     H.add(T₁)                              ▷ Single partition, max hash length
9:     **return** H
10:   // Calculate bigmin B and litmax L
11:   **if** cb is even **then**
12:     B, L ← calcBigminLitmax([T₁.X, T₂.X], [T₁.Y, T₂.Y], 1)
13:   **else**
14:     B, L ← calcBigminLitmax([T₁.Y, T₂.Y], [T₁.X, T₂.X], 0)
15:   rangeSearch(T₁, L, H)                                          ▷ split
16:   rangeSearch(B, T₂, H)                                          ▷ split
17:   **return** H

**Algorithm 2: Compute minimal set of z-order hashes.**

and numbers that may not intersect the queried interval but are in its neighborhood.

The range-based query predicates can be combined by adding or removing hashes to be searched in the index. For instance, if we would like to have a result set for a intersect query that only contains results that overlap without any subsumption, we can compute the set of hashes for containedBy and contains and remove them from the set of hashes for intersect. This can be computed by the query processing engine without the need to scan the index. The only parameters required are the overall ranges of temporal and numerical expressions in the document collection and the precision of the stored z-order hashes.

## 2.5 Query Optimizations

A naïve approach to process a query will be to perform a simple linear scan between pair of hashes corresponding to the query-MBRs. However, as noted earlier this may result in false positive results which can slow down the overall query processing. To reduce the number of spurious hits, we next describe three optimizations that can greatly speed up the overall query processing.

**Pruning Hashes for Invalid Subranges.** First, based on the uncertainty-aware model, we can prune away those search subranges where the begin of the interval is greater than the end i.e., $X > Y$. The hashes corresponding to these invalid subranges follow a recursive pattern. At the highest level of recursion, the entire bottom-right quadrant can be ignored along with two smaller triangles (see Figure 3). These triangles can be recursively broken down into smaller quadrants until the hashes covering the entire invalid area are found. The bottom-right quadrant is given by the hash $01_2$. Any value having this as its prefix is then invalid and can be ignored. The number of ignored values at each recursion level thus grows at a rate of $2^n$, where $n$ is the current recursion level (with the initial level at 0). These values can be pre-computed and stored in a lookup-table for pruning during query processing. The total number of ignored values for a given level of precision is then: $2^n + 2^{n-1} + 2^{n-2} + ... + 2^0 = \sum_{i=0}^{n} 2^{n-i} = 2^{1+n} - 1$.

**Adjusting Accuracy.** Second optimization that we perform is to reduce the precision of the z-order curve while still being within a minimum range accuracy. The accuracy of representing a single dimension on the z-order curve is given by Equation 4. Solving that equation for $n$ and rounding up to the nearest integer gives

```
Input: K and U :tuples for known dimension and d: indicator of unknown dimension.
Output: litMax and bigMin: calculated values for unknown dimension.
 1: procedure calcBigminLitmax(K,U,d)
 2:     cb ← number of common MSB between U[0] and U[1]
 3:     litMask ← bit-string equal to 011 . . . , total length equal cb
 4:     bigMask ← bit-string equal to 100 . . . , total length equal cb
 5:     litMask ← first cb MSBs of U[0] followed by litMask
 6:     bigMask ← first cb MSBs of U[1] followed by bigMask
 7:     if d = 0 then                              ▷ Unknown dimension is Y
 8:        litMax ← bit interleave litMask and K[1], starting wtih mask
 9:        bigMin ← bit interleave bigMask and K[0], starting with mask
10:     else                                       ▷ Unknown dimension is X
11:        litMax ← bit interleave K[1] and litMask, starting with K
12:        bigMin ← bit interleave K[0] and bigMask, starting with K
13:     return litMax, bigMin
```

**Algorithm 3: Computing BIGMIN and LITMAX values**

the minimum number of bits needed to be certain that any hashes calculated are not deviating more than the allowed maximum for that dimension. This can be restated as:

$$(e-b)/2^n \leq \text{Deviation} \tag{5}$$

As the range search is done over two dimensions, the calculated precision must be multiplied by two. If the dimensions differ in required accuracy, the precision required for the most accurate of the dimensions is used. The calculated number of bits is only the maximum precision needed and can be lowered for continuous hashes (discussed next). For larger ranges, potentially more hashes will be returned from the z-order curve, as the MBR may cover multiple quadrants at different precision levels. At the same time, searches over larger areas often require less precision to begin with. For example, if queried interval lies in the range of *[1900, 1999]*, having a potential ±10 year deviation in each end is not of great concern, compared to if the query was *[1990, 1999]*. This is utilized by defining the maximum allowed deviation as a part of the query processing. A larger range allows for a greater deviation, while a smaller range requires a smaller deviation. By dividing the smallest of the ranges by 10, a 10% deviation is achieved in the worst case. For instance, *[1900, 1999]* would then potentially allow for a ±10 year accuracy, while *[1990, 1999]* a ±1 year accuracy.

**Continuous z-order Values.** Third optimization that we perform is to convert the computed query-MBRs to a minimal set of hashes to query the inverted indexes. Specifically, we utilize a modified variant of BIGMIN and LITMAX method [34] that does not need to consult the inverted indexes for the minimal hash set computation. This method takes as input the overall search range depicted by the query-MBR (i.e., coordinates of the diagonally opposite corners of the MBR) and recursively splits the search space into sub-ranges verifying if z-order hash is continuous. The notion of continuity here conveys if the hash is relevant to the search range or not. The basis condition of recursion is reached when maximum precision is obtained for the z-order hash. To determine continuity we check if two hash values are continuous by determining the number of common MSBs using a bitwise XOR. Every pair of common bits in the count represents if the hash values lie in the same quadrant. By counting such pairs of common bits we can determine the common parent quadrant the hash values occupy. Having counted the common bits, a further verification is done to see if both the hash values are same (either all 0s or all 1s based on the corner of the diagonal) after the number of common bits to ensure they indeed lie in the same quadrant. Otherwise, this indicates a discontinuity

and a split must occur. These optimizations together are described in Algorithm 2. The third optimization of checking continuity is done on line 2. If they are continuous, the search can stop, and the common MSBs can be added to the return hash set (line 5). The second optimization is applied between lines 7-9. There the method checks if the number of common bits is equal to the maximum precision. If true, the search stops at the current recursion as the two hashes are seen as equal for the given precision, and the common bits are added to the return hashes. If none of the above terminating conditions are reached, then a discontinuity has been discovered and a split needs to be made (lines 10-14). The computation of this split is described next.

If a discontinuity occurs then the search space needs to be split into sub-ranges. To this end, we compute the split using a modified variant of BIGMIN and LITMAX method [34]. The split in the search space can either be horizontal or vertical. This is decided by the number of common bits between the pair of hashes describing the query-MBR. If there are an even number of common bits then the next bit which is not common is a Y-value (horizontal). Similarly, if there are an odd number of common bits then the next bit is an X-value (vertical). We can readily identify the values for one of the dimensions in the new split as they are already known and can be extracted from the current corners of the MBR. A horizontal split means that the search is split along the Y-axis and that the current X-values are within the search space. The BIGMIN and LITMAX calculation is then done to find the Y-values just to the left and to the right of this split. Similarly, a vertical split means a split along the X-axis, and the calculation is done to find the X-values just to the left and to the right of the split. The computation takes the values of the known dimension as one pair, and the values of the unknown dimension as another. The values are the extracted bit representation of each dimension of the hashes, i.e., starting with the MSB and taking every other bit yields the Y-value for that hash. The calculation differs based on which dimension the calculations is to be done for. A horizontal split would then take in both X-values as the known pair, and the Y-values as the unknown pair, with the computation being done for the Y-dimension.

The computation (shown in Algorithm 3) is straight-forward and requires appending a set of bits to the common MSBs of the values of the unknown dimension (lines 3-6). The common bits indicate where the dividing line of the split is. BIGMIN is the next (smallest) value within the search range in the upper part of the split. LITMAX is the largest value in the lower part of the split. BIGMIN's value for the unknown dimension is calculated by appending to the common bits. A 1 is appended to indicate the upper part of the split, followed by 0s such that it is the smallest value in this part. LITMAX's value for the unknown dimension is the inverse, appending a 0 followed by 1s. The final LITMAX and BIGMIN value is calculated by interleaving the values from the known dimension with the calculated LITMAX and BIGMIN values from the unknown dimension. LITMAX is interleaved with the largest of the known values, and BIGMIN with the smallest (lines 7-12). The starting value of the final hash is always the Y-value and a dimension bit indicator shows which dimension is the unknown and which is known (line 7).

In Figure 5a, we illustrate how the range search is divided into smaller sub-ranges. The two-dimensional range search is done from point S to point T equals a z-order range search from the

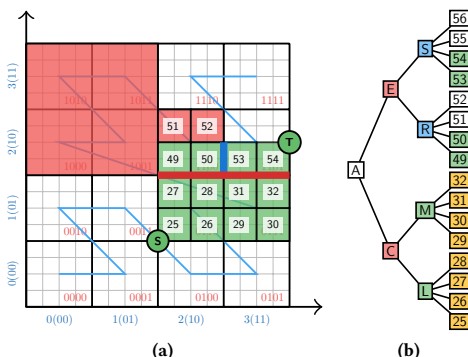

**Figure 5: (a) Example range search using the BIGMIN and LITMAX algorithm for finding continuous z-order hashes depicted using the z-order curve and (b) its corresponding quadtree.**

hash in quadrant 25 ($011000_2$) to the hash of quadrant 54 ($110101_2$), over a 6-bit z-order curve (only the four-bit hashes are shown). $S$ and $T$ are not continuous, as shown by the quadrants shaded red. The algorithm finds the number of common bits, which is zero, meaning a horizontal split and unknown Y-values. The extracted Y-bits from quadrant 25's hash is $010_2$ and from quadrant 54 $100_2$. As the current Y-values share no common bits, the Y-bits for BIGMIN (the BIGMIN mask) is simply $100_2$ and for LITMAX $011_2$. Interleaving $100_2$ with the (known) X-bits of quadrant 25 ($100_2$) results in a BIGMIN of $110000_2$, which is in quadrant 49. Similarly, interleaving $011_2$ with the X-values of quadrant 54 ($111_2$) yields a LITMAX of $011111_2$ (quadrant 32). Based on these values, the search range is split into two sub-ranges shown by the red line in Figure 5a. The new ranges are quadrant 25 to quadrant 32 and quadrant 49 to quadrant 54. The search from quadrant 25 ($01100_2$) to quadrant 32 ($011111_2$) is done in the same manner. However, this time the range is continuous. The hashes share three common bits, with the one diagonally opposite corner's hash having all zeroes following the common bits, and the other's hash having all ones. No further searching is then needed, and the hash ($011_2$) is added to the result hashes. The same does not apply for the search from 49 to 54, as the quadrants 51 and 52 are outside the search range, but between the z-order interval. The number of common bits between quadrants 49 and 54 is three ($110_2$), indicating a vertical split and unknown X-values. The X-bits of quadrant 49 are $100_2$ and $111_2$ for quadrant 54. They share one common bit, resulting in a BIGMIN mask of $110_2$ and a LITMAX mask of $101_2$. Interleaving the BIGMIN mask with the Y-bits of quadrant 49 ($100_2$) yields the BIGMIN value $110100_2$ (quadrant 53). Similarly, interleaving the LITMAX mask with the Y-values of quadrant 54 ($100_2$) yields $110001_2$ (quadrant 50). The new search ranges are then from quadrant 49 to 50 and from 53 to 54. The split is indicated by the blue vertical line in Figure 5a. Both these new ranges are found to be continuous as they share common bits, followed by all ones and all zeroes. The hashes added to the result are then 110002 and $11010_2$. However, NASH will only allow continuous ranges which have an even bit length, such that the prefix properties can be utilized in a base 4 encoding. This would result in two further vertical splits, and the hashes for 49, 50, 53, and 54 would be added individually. Figure 5b shows the different levels of the search in a quadtree. Only the visited subtrees are included. The red lines illustrate the first range split, and the blue

lines the second split. The green nodes indicate where the search terminated, and are the hashes added to the result set. The yellow nodes show the quadrants which were not visited as they were within a continuous range. As can be seen, a total of 12 leaf nodes are included in the search, but only 6 are needed to represent the entire range (four if allowing odd hash lengths). If allowing for some precision loss, even fewer hashes could be included in the results. For example, only the hashes for $E$ and $C$ could be included, however, it would come at a cost of decreased search accuracy.

**Table 1: Raw collection sizes and their annotation statistics.**

| COLLECTION | SIZE (GB) | #DOCUMENT | #WORD | #NE | #TIME | #NUMBER |
|---|---|---|---|---|---|---|
| **NYT** | 3.06 | 1.86 M | 1.06 B | 107.77 M | 15.41 M | 21.72 M |
| **WIKIPEDIA** | 33.44 | 6.34 M | 3.81 B | 626.27 M | 150.89 M | 115.14 M |
| **C4-NEWS** | 14.38 | 13.81 M | 6.14 B | 572.60 M | 85.18 M | 113.77 M |
| **C4-EN** | 304.64 | 365.07 M | 133.59 B | 8.36 B | 1.28 B | 2.17 B |

**Table 2: Index sizes for the document collections.**

| INDEX TYPE | NYT | WIKIPEDIA | C4-NEWS | C4-EN |
|---|---|---|---|---|
| N-GRAM | 44.6 GB | 152.7 GB | 283.5 GB | 2360.1 GB |
| ANNOTATION | 588.2 MB | 1.6 GB | 2.3 GB | – |
| Z-ORDER | 4.9 GB | 19.8 GB | 26.3 GB | 219.5 GB |
| DIRECT | 9.3 GB | 40.6 GB | 51.3 GB | 880.6 GB |

## 3 Evaluation

We next describe the evaluation setup for our experiments.

**Annotated Document Collections.** We instantiate indexes for four document collections. The first document collection we use is the New York Times Annotated corpus [7] which consists of news articles published the New York Times between the period 1987 to 2007. This document collection consists of approximately 1.8 million documents. The second document collection is the complete English Wikipedia [1], which consists of the collaboratively authored documents regarding notable entities and events. A recent snapshot of the entire English Wikipedia comprises of around 6.34 million documents. The third document collection we use is a news subset of the English Common Crawl, published as C4 [30]. This document collection consists of approximately 13.81 million documents. The fourth and final document collection we use is the entire C4-En corpus derived from the English Common Crawl. This is the largest collection in our evaluation setup amount to more than 365 million documents. We annotated each document collection using the Stanford's CoreNLP toolkit [25] for coarse-grained named entities and resolved temporal and numerical expressions. The temporal and numerical expressions were then mapped to our z-order based hashes. Complete statistics regarding the document collection statistics are given in Table 1.

**Implementation and Indexes.** We implemented the entire codebase for the indexing infrastructure from ground up in Java. The z-order hashes for temporal expressions were done with 40-bit precision so as to accommodate the hierarchical property of z-order curves. With 40-bit precision, we can query temporal expressions precisely at the day-level granularity. The z-order hashes for numerical expressions were done at 20-bit precision to accommodate large global numerical ranges. The z-order indexes store both the hashes of the temporal and numerical expressions along with their prefixes to leverage the hierarchical properties of the space-filling curves. In order to process event-oriented queries, we created a suite of indexes over the word layer, the coarse-grained named

entity layer, and the z-order hashes of the resolved values of temporal and numerical expressions as discussed in Section 2.3. Due to the size of the C4-En collection, we only instantiate unigram and bigram indexes for locating n-grams. Moreover, we omit the annotation only indexes for C4-En as the posting lists for a very small coarse-grained named entity vocabulary grow prohibitively long. Similarly, to reduce the size of C4-En direct index, we only store the word layer along with two other sentence-level annotations.

The document collections were pre-processed, annotated, and indexed on our Hadoop cluster of twenty five machines. Each machine in the cluster consists of the following minimal specifications: an Intel Xeon CPU with 16 cores and processing speed of 2.20 GHz and 128 GB of physical memory. The indexes are stored in HBase, a scalable distributed key-value store, which runs on our Hadoop cluster. HBase provides fault-tolerance and allows us to scale our query processing over millions of documents with ease. A summary of the indexes that we have created with their sizes are shown in Table 2. All our experiments were carried out on a compute node that is part of the Hadoop cluster.

**Query Testbed.** To construct a testbed of event-oriented queries, we extracted important historical event descriptions for each day of the year from the New York Times Learning Network — "On This Day" portal [5]. Each event-oriented query in the testbed contains combinations of annotations and the textual surface forms. For each query, we ensure that each query has at least one coarse-grained named entity and at least one temporal expression (i.e., its event date). In total, we have 4,548 queries with approximately 2.50 coarse-grained named entities (i.e., PERSON, LOCATION, ORGANIZATION, and MISC.) on average per query and approximately 1.54 temporal and numerical expressions on average per query. As an example, for the event: *"April 17, 1961, about 1,500 CIA-trained Cuban exiles launched an invasion at the Bay of Pigs on the southwestern coast of Cuba in a failed attempt to overthrow Fidel Castro"* [5]. The corresponding event query in the testbed is: {*(April 17, 1961)*⊕([1961-04-17 , 1961-04-17]), *(1,500)*⊕(∼1500.0), *(cuban)*⊕(MISC), *(bay of pigs)*⊕(LOC.), *(cuba)*⊕(LOC.), *(fidel castro)*⊕(PERSON)}. Where, ⊕ shows the combination of the word sequences corresponding to the named entity annotations.

**Setup.** We evaluate our system and the baselines for end-to-end query processing runtimes. To do so, we sample 100 queries from the testbed and run them three times with cold-cache setup. Cold-caches are simulated by shuffling the queries in between rounds. We report here results based on cold-cache settings as the results are similar to that of warm-cache setup. We issue each event-oriented query along with their event date. We evaluate each query in the sample as a conjunctive query, i.e., the matched documents must contain all the coarse-grained named entities with its event date and accompanying numerical and temporal expressions. We further evaluate the effect the query optimizations have on NASH by varying the search accuracy by 5%, 10%, and 25%.

**Baselines.** We first establish a naïve baseline of the total time required to perform a distributed scan over the document collections on our Hadoop cluster. With the SCAN baseline, we obtain a worst case bound on query runtimes. We further compare our system to two additional baselines: TEXTI and ANNI. The TEXTI baseline processes a query by first identifying a candidate set of documents

**Table 3: Indexes used by the baselines and NASH.**

| SYSTEM | N-GRAM | ANNOTATION | Z-ORDER | DIRECT |
|---|---|---|---|---|
| SCAN | × | × | × | × |
| TEXTI | • | × | × | • |
| ANNI | • | • | × | • |
| NASH | • | × | • | × |
| NASH-OPT | • | × | • | × |

**Table 4: Runtimes in seconds for a distributed scan.**

| NYT | WIKIPEDIA | C4-NEWS | C4-EN |
|---|---|---|---|
| 76 | 330 | 291 | 5698 |

corresponding to the event coarse-grained named entities by retrieving the results using their phrases from the word layer. The TEXTI baseline then ensures that the candidate set of documents indeed contain the event date and accompanying temporal and numerical values by referring to the resolved annotation layers for each of the candidate documents by using the direct index. The ANNI baseline, works similar to the TEXTI baseline but differs in how it matches the temporal and numerical expressions. The ANNI baseline, identifies potential matches for the event date and accompanying temporal and numerical expressions by using an inverted index over coarse-grained named entity annotation layer. Having identified the potential matches, it ensures that the resolved values for the coarse-grained named entities are indeed correct using the direct index. Note that the ANNI baseline is functionally similar to a baseline utilizing indexes over numerical intervals. This baseline would be slower than ANNI as it would entail scanning the interval indexes multiple times for the range search predicates. For the C4-En collection, evaluating the baselines TEXTI and ANNI is prohibitively expensive as it may entail scanning the entire document collection using the direct index. Thus, for the C4-En collection we provide comparative results using the distributed scan baseline (SCAN).

We evaluate two variations of our proposed system: NASH and NASH-OPT. Both variations of our proposed system, NASH and NASH-OPT, utilize only the z-order based indexes to match the event dates and the accompanying temporal and numerical expressions. For NASH we disable the query optimizations and ensure that sets of hashes generated are at full precision for retrieval of temporal and numerical expressions. While for NASH-OPT we leverage the query optimizations that further adjust the accuracy (deviation set to 1%) during query processing. We evaluate both the baselines and our system for all four of the range-based search predicates: `contains`, `containedBy`, `intersect`, and `near` (Δ = 0). A summary of the indexes the systems utilize is given in Table 3. As the baselines, TEXTI and ANNI, utilize the direct indexes we can ensure that the results sets obtained are equivalent to NASH. However, when applying the additional optimizations during query processing for NASH-OPT, e.g., adjusting accuracy, we may obtain additional matches that contain temporal and numerical ranges in the proximity of queried intervals.

**Distributed Scan Results** for each of the four document collections are shown in Table 4. As can be seen the time taken for scanning each collections is proportional to its size. For Wikipedia the time taken is more than C4-News (even though it is larger than Wikipedia). This is because Wikipedia documents are longer than the documents contained in C4-News archive. The average number of words per document for Wikipedia is approximately 600.95 words as compared to 444.61 words per document for C4-News. For New York Times, Wikipedia, and C4-News the distributed scan over our Hadoop cluster can be done in the order of several minutes. For the Web-scale English C4-En collection, the distributed scan

**Table 5: End-to-end runtimes (in seconds) for cold caches.**

| RUNTIME RESULTS FOR PREDICATE containedBy. | | | |
|---|---|---|---|
| **SYSTEM** | **NYT** | **WIKIPEDIA** | **C4-NEWS** | **C4-EN** |
| TEXTI | 51.25 ± 220.94 | 175.02 ± 603.75 | 182.03 ± 647.92 | – |
| ANNI | 47.21 ± 192.25 | 202.92 ± 562.58 | 224.03 ± 580.49 | – |
| NASH | 2.87 ± 2.53 | 62.69 ± 52.13 | 16.54 ± 15.29 | 235.27 ± 267.74 |
| NASH-OPT | 2.29 ± 3.25 | 11.56 ± 10.18 | 10.86 ± 13.66 | 202.93 ± 242.54 |

| RUNTIME RESULTS FOR PREDICATE contains. | | | |
|---|---|---|---|
| **SYSTEM** | **NYT** | **WIKIPEDIA** | **C4-NEWS** | **C4-EN** |
| TEXTI | 46.17 ± 204.68 | 170.52 ± 587.82 | 177.86 ± 628.58 | – |
| ANNI | 47.77 ± 194.30 | 202.35 ± 561.09 | 222.37 ± 578.65 | – |
| NASH | 0.90 ± 1.14 | 4.85 ± 5.26 | 6.16 ± 9.61 | 122.81 ± 174.81 |
| NASH-OPT | 0.77 ± 1.06 | 4.11 ± 4.61 | 6.02 ± 9.49 | 121.81 ± 177.89 |

| RUNTIME RESULTS FOR PREDICATE intersect. | | | |
|---|---|---|---|
| **SYSTEM** | **NYT** | **WIKIPEDIA** | **C4-NEWS** | **C4-EN** |
| TEXTI | 45.75 ± 203.90 | 170.39 ± 587.65 | 176.07 ± 622.11 | – |
| ANNI | 47.30 ± 193.24 | 201.70 ± 554.79 | 224.21 ± 580.00 | – |
| NASH | 2.96 ± 2.78 | 62.82 ± 52.76 | 16.99 ± 16.63 | 245.87 ± 299.20 |
| NASH-OPT | 2.53 ± 3.69 | 12.00 ± 10.82 | 10.78 ± 13.38 | 206.93 ± 245.22 |

| RUNTIME RESULTS FOR PREDICATE near. | | | |
|---|---|---|---|
| **SYSTEM** | **NYT** | **WIKIPEDIA** | **C4-NEWS** | **C4-EN** |
| TEXTI | 45.77 ± 202.45 | 170.14 ± 586.16 | 176.11 ± 622.89 | – |
| ANNI | 47.44 ± 192.75 | 201.16 ± 555.28 | 222.54 ± 580.13 | – |
| NASH | 1.11 ± 1.59 | 5.72 ± 6.59 | 9.16 ± 15.91 | 130.78 ± 182.41 |
| NASH-OPT | 1.02 ± 1.54 | 4.98 ± 6.06 | 9.44 ± 16.67 | 130.52 ± 186.24 |

**Table 6: End-to-end runtimes (in seconds) by adjusting accuracy.**

| RUNTIME RESULTS FOR PREDICATE containedBy USING NASH-OPT. | | | |
|---|---|---|---|
| **DEVIATION** | **NYT** | **WIKIPEDIA** | **C4-NEWS** | **C4-EN** |
| 5 % | 2.29 ± 3.35 | 11.36 ± 10.14 | 10.74 ± 13.69 | 202.59 ± 242.81 |
| 10 % | 2.65 ± 3.88 | 12.19 ± 10.10 | 11.16 ± 13.48 | 205.31 ± 239.92 |
| 25 % | 2.47 ± 3.34 | 14.47 ± 12.57 | 18.35 ± 22.30 | 307.89 ± 330.17 |

| RUNTIME RESULTS FOR PREDICATE contains USING NASH-OPT. | | | |
|---|---|---|---|
| **DEVIATION** | **NYT** | **WIKIPEDIA** | **C4-NEWS** | **C4-EN** |
| 5 % | 0.78 ± 1.14 | 4.04 ± 4.53 | 6.26 ± 10.13 | 119.72 ± 175.30 |
| 10 % | 0.77 ± 1.09 | 4.00 ± 4.39 | 6.07 ± 9.62 | 118.95 ± 174.43 |
| 25 % | 0.68 ± 0.94 | 3.64 ± 3.92 | 5.54 ± 8.59 | 120.07 ± 176.46 |

| RUNTIME RESULTS FOR PREDICATE intersect USING NASH-OPT. | | | |
|---|---|---|---|
| **DEVIATION** | **NYT** | **WIKIPEDIA** | **C4-NEWS** | **C4-EN** |
| 5 % | 2.58 ± 3.81 | 11.49 ± 10.31 | 11.24 ± 14.25 | 205.39 ± 244.61 |
| 10 % | 2.51 ± 3.70 | 12.06 ± 10.06 | 11.57 ± 13.80 | 207.77 ± 240.88 |
| 25 % | 2.59 ± 3.59 | 14.79 ± 12.90 | 18.24 ± 21.83 | 308.02 ± 328.19 |

| RUNTIME RESULTS FOR PREDICATE near USING NASH-OPT. | | | |
|---|---|---|---|
| **DEVIATION** | **NYT** | **WIKIPEDIA** | **C4-NEWS** | **C4-EN** |
| 5 % | 0.99 ± 1.53 | 4.91 ± 6.01 | 9.46 ± 16.67 | 127.23 ± 182.80 |
| 10 % | 0.85 ± 1.34 | 4.85 ± 5.83 | 9.19 ± 16.09 | 124.98 ± 179.04 |
| 25 % | 0.81 ± 1.19 | 4.13 ± 4.58 | 6.93 ± 11.25 | 124.81 ± 180.03 |

takes the longest: approximately 1 hour and 35 minutes. When comparing the results of a distributed scan to that of the baselines (TEXTI and ANNI) in Table 5, we observe that their runtimes are in the same neighborhood of minutes for New York Times, Wikipedia, and C4-News document collections.

**End-to-End Runtime Results** of our experiments are reported in Table 5. We first discuss the results for the range-based query predicates containedBy and intersect. These predicates are challenging for NASH to compute as they require computing query-MBRs for temporal and numerical expressions that can subsume (containedBy) or intersect the queried intervals. The results for the range-based query predicates containedBy and intersect show our system NASH outperforms the TEXTI baseline by a factor in the range of 15.46 - 17.86× for the New York Times document collection. For Wikipedia, which contains the most number of temporal and numerical expressions, and C4-News our system outperforms the most competitive baseline TEXTI by a factor in the range of 2.71 - 11.01×. This speedup can be attributed to the fact that the TEXTI baseline needs to consult the direct index for each document in the candidate result set to ensure that the predicate conditions of each of the range-based search operations are satisfied.

When comparing our system NASH to ANNI baseline our system performs 3.24 - 16.45× faster for containedBy queries and 3.21 - 15.98× faster for intersect queries. This speedup can be attributed to smaller posting lists corresponding to the z-order hashes as compared to lengthier posting lists corresponding to coarse-grained named entities types of NUMBER and DATE. Therefore, it takes longer for the ANNI baseline to process a conjunctive event-oriented query. Whereas, for NASH multiple hashes corresponding to the query-MBR can help prune irrelevant documents early in the query processing. When further adjusting for accuracy, we can further lower our runtimes with NASH-OPT and obtain speedups in the range of 15.14 - 22.38× for containedBy predicate and 14.20 - 20.80 × for intersect predicate. Also, we observe that NASH-OPT significantly reduces the query runtimes for Wikipedia, which contains the maximum average number of temporal and numerical expressions per document.

We next discuss the results for the range-based query predicates contains and near. These predicates are more probable to be requested by users such as journalists and scholars in digital humanities as they enable a proximity and containment search around a query interval. The results for the range-based query predicates contains and near show that NASH, provides speedups in the range of 41.23 - 51.30× when compared to the TEXTI baseline for the New York Times collection. For Wikipedia and C4-News, we obtain speedups in the range of 28.87 - 35.16× for contains queries and speedups in the range of 19.23 - 29.75× for near queries. When comparing our system NASH to ANNI baseline, our system performs 36.10 - 53.10× faster for contains queries and 24.30 - 42.74× faster for near queries. Again, the speedup in this case can be attributed to the larger posting lists associated with coarse-grained named entities when compared to the posting lists for z-order indexes. Overall, for NASH we observe that it is quicker to execute the queries corresponding the predicates contains and near when compared to executing same queries with the predicates containedBy and intersect. This is because the number of hashes needed to describe the query-MBRs for contains and near queries are fewer than those needed to describe containedBy and intersect. When further adjusting for accuracy using NASH-OPT we observe a less significant improvement. This can be attributed to the observation that a set of more precise (longer) hashes are required for proximity queries when compared to intersection queries and therefore are least affected by the adjustment for accuracy. Overall, for NASH-OPT, we see speedups in the range of 29.55 - 62.04× for contains and 18.66 - 46.51× for near queries.

For the C4-En document collection, we draw comparative results based on the SCAN baseline. As we have seen before, the SCAN baseline is competitive for the task of range search and performs similarly when comparing to TEXTI and ANNI that leverage n-gram and annotation indexes for the smaller three document collections. We observe that NASH provides a speedup in the order of 23.18 - 24.22× for the containedBy and intersect query predicates over SCAN. While NASH-OPT provides a speedup in the range of 27.54 - 28.08× for the same query predicates. For the query predicates contains and near, NASH provides speedups in the range of 43.57 - 46.40× over SCAN. Whereas NASH-OPT provides speedups in the range of 43.66 - 46.78× for contains and near query predicates. These speedups are similar to the speedups for other collections in

our evaluation setup and thus demonstrate that our system NASH can reliably scale to the Web for event-analytics.

**End-to-End Runtime Results by Adjusting Accuracy.** We further evaluate NASH-OPT for end-to-end query execution by adjusting for accuracy (see Table 6). To that end, we execute the same sample of 100 queries in cold cache setup while setting deviations of 5%, 10%, and 25% in search accuracy of the ranges (see Equation 5). We first discuss the results for the query predicates `containedBy` and `intersect`. We observe that as we increase the deviation, the runtimes overall increase across the document collections. We see the most significant increase in runtimes when executing the queries with deviation at 25%. The trend of increasing runtimes with increasing deviation percentages can be attributed to shorter hashes that describe the query-MBRs. As discussed in Section 2.5, shorter hashes correspond to larger search regions that intersect a larger number of temporal and numerical expressions when compared to longer hashes (indicative of higher search accuracy) that describe a precise search region referring to only the required intervals. This is because they describe a larger search region that subsume and intersect the query interval (see Figure 3c). Since, the query predicates `containedBy` and `intersect` require looking up of such shorter hashes as compared to the query predicates `contains` and `near` they consume more time as the deviation is increased.

We next discuss the results for the query predicates `contains` and `near`. For these query predicates, we observe that as we increase the deviation percentage the overall query execution time decreases marginally. The insignificant decrease query runtimes can be attributed to three factors. First, `contains` and `near` query predicates implicitly require longer hashes to describe the neighborhood of the search region and thus are least affected by the reduction in the search accuracy. Second, for the proximity-based queries since the search region is described by longer hashes it entails lookup of shorter posting lists of temporal and numerical expressions. Third and finally, the deviation in search accuracy has a more pronounced effect when indexing temporal and numerical hashes with a higher-precision z-order curve. Such a requirement may arise when indexing financial documents containing a higher number of numerical expressions located closer to to each other in the search space. For Web search, where temporal and numerical expressions are less densely located in the search space a lower-precision z-order curve is sufficient and may provide reliable performance.

**Index Sizes.** NASH utilizes z-order indexes that are a factor of at least 1.90× and at most 2.05× smaller than storing the direct index. For extracting text regions corresponding for event-oriented queries, we need to only maintain a smaller direct index that records only the word layer. For instance, the C4-En z-order indexes are 4.01× smaller than the direct index, where fewer layers are stored.

## 4 Related Work

Earliest works in search and indexing using space-filling curves are [23, 34]. Tropf and Herzog [34] showed that query processing using z-order curves results in a logarithmic speed up with the number of indexed records. Lawder and King [23] demonstrated methods for utilizing space-filling curves for multi-dimensional indexing. Their approach, which is similar to the BIGMIN and LITMAX method [34], utilized Hilbert curves to implement a search infrastructure with three dimensions.

Space-filling curves have found wide applicability for indexing geo-spatial data as demonstrated in several works and publicly available systems [2–4, 15, 20, 24, 27, 35, 37]. Examples of systems leveraging space-filling curves are GeoWave [35] and MD-HBase [27]. Both GeoWave and MD-HBase utilize space-filling curves to index geo-spatial data over distributed key-value stores to speed up query processing. GEOHASH.ORG [2] is a publicly available API that allows users to represent latitude and longitude pairs into base-32 z-order hashes (geohashes). Similar approaches have been widely adopted in open-source NoSQL database systems such as ElasticSearch [3] and MongoDB [4]. A more recent work [24] combines both these threads of research. Lee et al. [24] show how to index geo-spatial data using geohashes in HBase such that it can support the fundamental spatial query operations. The authors are able to perform these spatial operations by leveraging the hierarchical characteristics that geohashses implicitly model. Prior works on combining spatial and text search has resulted in approaches that combine spatial indexing methods such as R-Trees with inverted indexes over words [15]. However, such approaches are less flexible and not scalable as there is redundant storage of location-based data for each word in the collection. Similarly, prior works such as [37] present approaches that do not support complete set of range-based operations and rely only on efficient top-k query processing based on proximity scores.

Earlier works on range-based search in uncertain metric-spaces [16], interval-based data [9], and temporal databases [22] all rely on $B^+$-trees as their underlying data structure which can not immediately be utilized in large-scale search infrastructures that primarily rely on inverted indexes for query processing. In [12], the authors index versioned document collections, where versions correspond to timestamps. They utilize a partition-based approach to split posting lists corresponding to different timestamps enabling quick execution of timepoint-based queries. While in [8], the authors propose a query optimization framework to extend querying to time intervals with an additional I/O constraint to retrieve a subset of the relevant result set using hash-based KMV synopsis. A comprehensive survey of semantic search using text and knowledge graphs [10] shows that most storage techniques for annotated document collections ignore to support advanced capabilities for document retrieval that contain temporal and numerical expressions. Even modern transformer-based large language models fall short of learning semantics of numbers present in large document collections [31].

## 5 Conclusion

We described NASH, a search system that provides the capability to perform range-based search over intervals that can be implicitly contained within natural language expressions of time and numbers. To do so, NASH utilizes z-order curves that enable us to search for intervals at different levels of granularity. Furthermore, we described the concept of continuous hashes and an improved optimization based on the BIGMIN and LITMAX approach to speedup the query processing. Our results show that NASH provides impressive speedups in the order of 19.23 - 53.10× for more probable `contains` and `near` queries. Moreover, NASH provides these speedups by utilizing z-order indexes that are 1.90 - 2.05× smaller than the direct index required otherwise.

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
