# OpenReview forum: "Scalable Range Search over Temporal and Numerical Expressions"
_ACM.org/SIGIR/ICTIR/2024/Conference — ICTIR 2024_

### Official Review · Reviewer_7xh4 · 2024-05-03

**Rating:** -2
**Confidence:** 4

**Objective Part Of Review:**

This paper describes a solution for combining numeric range search with text retrieval. As index structure, z-metric space filling curves in combination with quad trees are proposed, a spatial access method well known in the database field, whose origins can be traced back to the 1980s [1]
The paper describes the basic techniques and query optimization methods used, one of which seems to be new. Experiments with three data sets and synthetic queries show substantial improvements in terms of efficiency.
However, relevance does not play any role here, as only exact matches (for numeric ranges) are considered, so also retrieval quality is not an issue here.

1. GABOW, Harold N.; BENTLEY, Jon Louis; TARJAN, Robert E. Scaling and related techniques for geometry problems. In: Proceedings of the sixteenth annual ACM symposium on Theory of computing. 1984. S. 135-143.

**Subjective Part Of Review:**

This paper takes a typical database approach to the problem of combining exact text match with range search, as relevance and ranking are not an issue here. Spatial data structures (for which range search and intervals are the most simple case) have been investigated in the database community since the mid-1980s. Unfortunately, it does not become clear where this paper adds anything new to this area (besides the improvement of an older optimization method).
Today's experimental data sets may be bigger and the information extraction methods are substantially better than in the last century, but apart from having better test collections, I cannot see any scientific progress in this paper.

---

### Official Review · Reviewer_A1V3 · 2024-05-08

**Rating:** 2
**Confidence:** 4

**Objective Part Of Review:**

In this paper the authors describe their use of space filling curved for the encoding and search of numerical ranges in text data.  The numerical ranges are normally dates, but other numbers are covered too.

The encoding using z-order curves is very clever and allows the authors to search at an arbitrary granularity.  However, as I understand the paper, it’s still necessary to encode the entire numeric range at the lowest level – so that contains queries can be performed.  Such a case might be a numeric range of 0 to 1 billion matching a query of “contains 56-96”.  The authors should clarify this point as it could easily lead to an enormous vocabulary when indexing.

Citations are not nouns, please refrain from “In [12]” and similar uses as if they are.

Typos

Page 6, 110002 -> 11000_2

Page 89, to to -> to

**Subjective Part Of Review:**

The paper is well written and easy to follow.

---

### Official Review · Reviewer_em64 · 2024-05-13

**Rating:** 0
**Confidence:** 3

**Objective Part Of Review:**

Using two-dimensional spatial indexing to handle temporal and numerical data is novel and well-justified. The evaluation covers multiple document collections and a variety of query types, however the paper focuses mostly on system-centric metrics such as speedup and index size. Including user-centric evaluation metrics (P@k, NDCG@k) would provide a better insight of the system's performance.

While the system outperforms the defined baselines, the comparison against more standard IR indexing system is lacking. A comparison with systems like Lucene/Anserini can provide better insight about the current research landscape.

**Subjective Part Of Review:**

The paper is well-written and well-organised. The topic is interesting and related to ICTIR.

---

### Official Review · Reviewer_LFQN · 2024-05-17

**Rating:** 1
**Confidence:** 4

**Objective Part Of Review:**

The paper proposes a system that allows to combine entity and keyword based search with operators optimized for temporal and interval expressions. The paper reuses existing methods for extracting uncertain temporal and numeric expressions from text and for representing the resulting uncertain intervals. To allow for an efficient retrieval based on several search operators, the paper applies z-order codes, together with several novel optimizations. This is a novel combination of existing methods.

The paper is easy to read. It provides sufficient examples to understand why the proposed operators are useful and how the proposed index operates. The provided figures are useful. The paper provides a formally sound description of its techniques. The focus is clearly on system aspects, which include how indexes are organized and how they are applied during search.

For z-code hashes, the paper assumes that the largest numerical value that occurs in the collection is known. There are two potential issues here. First, the largest numerical value could be really huge (assume a document talks about the number of atoms in the universe, for example). It seems that this design choice makes all z-code hashes very long. Second, what happens when the collection changes and new (i.e., larger) numbers are added, does this mean that the z-code hashes need to be recomputed? The paper should probably discuss these issues.

**Subjective Part Of Review:**

The paper clearly considers an important and underexplored topic. Efficiently managing temporal and numeric expressions in context of search engines is vital for certain applications, and the paper provides some useful examples for information needs that would benefit from temporal query operators. The paper builds on existing work in terms of the annotation of temporal expressions, the underlying model for representing uncertain temporal expressions, and z-order curves for indexing arbitrary intervals and points. The actual contribution of the paper is the resulting system, which comes with well-designed indexes that smoothly integrate temporal operators with entity and keyword based query expressions. The paper cleverly explores options for improving standard z-order hashes, which helps to further optimize query processing.

The experimental evaluation is adequate, with four collections of different size (both in terms of number of documents and size of documents) and content. Experiments are done with only 100 queries, which is not very much for experiments regarding efficiency where the possible queries are not constrained by the availability of relevance assessments. It is unclear why the paper initially reports 4,548 queries, out of which only 100 are used for the actual experiments. Baselines are reasonable; the scan-based method is clearly a very naive baseline that would usually not be applied within the context of a search system, but gives an idea of worst-case times.

The demonstrated query processing times are impressive, especially as they were not achieved with a highly tuned in-memory framework. Response times of a few seconds for such large collections are fully adequate for most applications. Compared to the performance of NASH alone, the additional improvements with the optimizations are rather small.

The paper does not report very well how the actual implementation was done. It reports some details regarding the Hadoop-based implementation and the HBase-based index storage, but more details regarding how indexes were represented in HBase would have been useful. The paper would clearly be more valuable if the source code for the system, ideally together with queries and detailed experimental results, was made available.

The paper clearly puts its focus on the NASH system, and as a system paper, I consider it useful and interesting. It is not so clear to me, however, how it matches the more theory-oriented theme of ICTIR, especially as the contributions are on the systems side and not so much on the underlying model (which existed before and are reused). This is the main weakness of an otherwise good paper.

---

### Meta-Review · Area_Chair_A19w · 2024-05-24

**Recommendation:** Accept (Oral)
**Confidence:** 4

**Metareview:**

The paper proposes the combination of keyword-based search with the search for temporal and numeric interval expressions. It applies existing methods for resolving temporal/numeric expressions from text, and uses z-order hash indexes for supporting search with basis on temporal/numeric restrictions (which are seen as intervals in a 2D Cartesian space).

The innovative contributions relate to optimizations in query processing with z-order indexes, and to the combination of z-order hashes with functionalities for keyword-based search (i.e., both the z-order hashes and the keywords are stored in a common index associated to inverted lists), aiming at high speedups in query processing, and small index size.

The methods advanced in the paper are perhaps closer to the area of "databases", instead of "information retrieval", as for instance "relevance" and "ranked retrieval" are not considered as part of the evaluation criteria. However, I also believe that this particular aspect of relevance can be somewhat subjective, and this meta-review mostly attempts to follow the objective comments supported by falsifiable evidence. Still, with basis on the somewhat limited depth of the novel contributions that are provided, I am recommending that this manuscript can perhaps be accepted as a poster.

The experimental results are interesting, but ideally the authors should consider more queries in the tests (which should be easy, since there is no need to collect relevance judgements in order to measure speedups) and also comparisons against other approaches (e.g., in the databases literature, there are actually several proposals of data structures to simultaneously index/search over 2D intervals and text).

Reviewer LFQN provides relevant and interesting feedback that can be considered for improving the manuscript, and reviewers 7xh4 and em64 make good points about how the proposed contributions should have a stronger relation to the general area of "information retrieval."